



# Restoration of the Baltic Proper by decadal oxygenation of the deepwater

Anders Stigebrandt

Dept of Marine Sciences, University of Gothenburg, Box 460, SE-40530 Gothenburg, Sweden

*Correspondence to:* Anders Stigebrandt (anders.stigebrandt@marine.gu.se)

**Abstract.** Sediment core data from the Baltic Proper show that deepwater sediments during the present brackish state have alternated between anoxic and oxygenated episodes, implying that anoxic episodes were shut off by a natural restoration mechanism. This mechanism is identified as sustained oxygenation of the deep bottoms which shuts off the internal phosphorus (P) source from anoxic sediments. When this happens, P sink processes rapidly reduce the winter surface concentration $c_1$ of P. A P budget model shows that the presently eutrophic Baltic Proper can be restored in about 10 years by man-made (or natural) sustained oxygenation of the deepwater. This will reduce the total P supply and $c_1$ to about 25 % of present day values and the oxygen demand in the deepwater will be reduced proportionally, meaning that the natural water exchange again should be able to oxygenate the Baltic Proper and the restoration equipment can be removed.

## 1 Introduction

Sediment core data demonstrate that deepwater sediments in the Baltic Proper have switched between oxygenated and anoxic episodes during the present several-thousand-years-long brackish water phase. During anoxic episodes there were simultaneous intense cyanobacterial blooms that likely were fueled by phosphorus (P) released from anoxic sediments (Bianchi et al. 2000; Zillén and Conley 2010; Jilbert and Slomp 2013). Applying a P budget model, this internal P source was estimated to 2.3 g P m$^{-2}$ yr$^{-1}$ using contemporary data from the Baltic Proper (Stigebrandt et al. 2014). In addition to the

internal source the model also considers the external source, from land run-off and point sources from human activities, and the Kattegat source due to inflow of deepwater from Kattegat.

Anoxic episodes might have been initiated by strong salinity stratification in the deepwater (Bianchi et al., 2000), which may cause prolonged residence time for the deepest deepwater during which oxygen may be exhausted even if the supply of

organic matter produced in the surface layer is quite small. The residence time is then longer than the time it takes to consume the oxygen brought in by new deepwater. Episodes of strong salinity stratification in the Baltic Proper deepwater are consequences of the huge variability of the salinity of new deepwater that is a mixture between North Sea water (salinity ≈ 35) and surface water from the Baltic Proper (salinity ≈ 8), as demonstrated by a heuristic mechanical model (Stigebrandt et al. 2015a). Such episodes may have long duration because the rate of salinity reduction in the deepwater is slow due to

weak vertical mixing.



During periods with anoxic bottoms, the internal P source is turned on and the amount of P in the anoxic deepwater increases. Vertical circulation transports dissolved P from the deepwater to the surface layer. This eventually leads to increased amount of P and increased production of organic matter in the surface layer, which increases the vertical export of organic matter, which increases the oxygen consumption in the deepwater, which increases the area of anoxic bottoms and thereby the internal P source. This positive feedback loop tends to prolong periods of anoxia.

The Baltic Proper obviously possesses a mechanism to restore itself from eutrophication. This may be concluded from sediment core data showing that anoxic episodes were shut off and replaced by oxygenated episodes. This natural mechanism of restoration from eutrophication has not earlier caught much interest in the Baltic Sea literature. However, a high resolution sediment study shows that terminations of anoxic periods have been rapid (Jilbert and Slomp, 2013). The crucial quality of a restoration mechanism is that it may shut off the internal P source during a sufficiently long time during which the P content in the water column will decrease due to the action of P sinks, mainly in oxic bottoms. Oxygenation of the deep bottoms can shut off the internal P source (Stigebrandt et al. 2014). The time needed for recovery should depend on the magnitude of the external P supply which was small during the 19[th] century and earlier (Larsson et al. 1985) although it has been suggested that it might have been appreciable during certain phases of the medieval (Zillén and Conley, 2010).

Efficient oxygenation of the deepwater may occur when the salinity of new deepwater is low and varies little in extended periods. Such periods are likely due to the huge variability of the salinity of new deepwater. In this low-salinity mode of deepwater renewal, the deepwater stratification weakens and the deepwater volume decreases due to erosion by vertical mixing in the surface layer that extends down to the halocline in autumn and winter. A long episode of low-salinity deepwater renewal occurred in the 1980s and the beginning of the 1990s during which the top of the halocline was lowered from its usual level at about 60 m to about 100 m depth and the P content decreased in the Baltic Proper (Stigebrandt and Gustafsson, 2007). Anoxic deepwater almost disappeared and the volume of hypoxic deepwater was halved (Fig. 1). However, this spontaneous restoration attempt by Nature was not completed when the deepwater again became more stratified by a major inflow of new deepwater in 1993. A proposal to apply sustained man-made oxygenation as a method to defeat eutrophication of the Baltic Proper was inspired by this episode (Stigebrandt and Gustafsson, 2007). Periods of deepwater renewal in the low-salinity mode, recognized as periods with lowered halocline, occurs rarely as can be seen from hydrographical data starting in the 1890s presented by Carstensen et al. (2014).

From the 1890s, when regular oxygen observations started, to the beginning of the 1950s there were only small oxygen-free areas in the Baltic Proper (Savchuk et al. 2008; Carstensen et al. 2014). Thereafter, anoxic bottoms expanded quickly, probably promoted by increased external supply of P. However, the strongly increased vertical stratification in the deepwater, induced by an extremely large and salty inflow of new deepwater in 1951, might have triggered the expansion of



anoxia. Decreased oxygen solubility due to increased temperatures during the last century may have worsened hypoxia and anoxia (Carstensen et al. 2014).

Due to increased sewage treatment and improved agricultural practice, the external phosphorus (P) supply to the Baltic proper decreased by about 50% from the peak in the 1980s to present time (Gustafsson et al., 2012). Despite this reduction, the phosphorus content in the water column of the Baltic proper increased by about 20% during the same period (Stigebrandt et al. 2014). This explains the increased eutrophication and the increased volume of anoxic water occurring after the ending of the spontaneous restoration attempt in 1993 (Fig. 1). The increasing phosphorus content in the water column can be explained by an increasing internal source of phosphorus, which more than well compensated for the decreasing external P supply (Stigebrandt et al., 2014).

The P budget model of the Baltic Proper developed in Stigebrandt et al. (2014) is here used to compute the equilibrium (steady state) winter surface layer concentration $c_1$ of P as function of the total P supply, the latter equals the sum of the external, and the internal, and the Kattegat supplies. The model is described in Section 2 below. Thereafter follows in Section 3 presentation and discussion of model results. In Section 4 a possible restoration of the Baltic Proper is discussed and the paper is brought to an end by a concluding discussion in Section 5.

## 2 Materials and methods

Inspired by lake modeling presented by Vollenweider (1969), an elementary nutrient budget model was developed and applied to the Baltic Sea (Wulff and Stigebrandt 1989). From known external supplies and changed storage in the water column, the internal and external (by water export) sinks of nutrients were computed. Predictions were made of future nutrient states of the Baltic Sea for various nutrient loading scenarios. According to that model, the winter phosphorus concentration in the surface layer should decrease when the external load decreases. However, as already mentioned, the winter surface concentration actually increased by about 20% although the external supply was halved during the period 1980 – 2005 (Stigebrandt et al. 2014). This shows that the model lacks a major phosphorus source. Therefore an internal source proportional to the area of anoxic bottoms was introduced in an extended P model (Stigebrandt et al. 2014).

In the present paper, I use the extended P model to compute the steady state (equilibrium) winter surface water P concentration in the Baltic proper that should be attained at the end of a restoration effort where the internal source is shut off. The model has two layers (Fig. 2). The upper layer is vertically mixed each winter down to the halocline whereby water from the lower layer is entrained into the surface layer. The lower layer is vertically stratified. For the phosphorus model the layers are added to form one single layer. Hereby the internal dynamics between the layers vanish from the model and only



fluxes perpendicular to external boundaries, including the sea bed, remain. However, when discussing oxygen conditions in the deepwater, the conceptual two-layer model is needed, see Sec. 3.

The time dependent equation (time resolution 1 year) for the total content of phosphorus in the water column of the Baltic proper reads

$$V\frac{d\overline{c}}{dt} = Extsource + Intsource + Q_1 c_0 - (Q_f + Q_1)\overline{c}_1 - Int\sin k \qquad (1)$$

Here $V$ is the volume of the Baltic Proper and $\overline{c}$ the volume mean winter concentration of phosphorus so that $V\overline{c}$ is the total winter content of P in the water column. $\overline{c}_1$ is the annual mean concentration in the surface layer and $c_0$ the time mean

concentration of phosphorus in the water flowing into the Baltic from the entrance area. These concentrations are important for the exchange (export and import) of phosphorus with Kattegat. It is assumed that $\overline{c}_1 = \gamma c_1$ where $c_1$ is the winter concentration of phosphorus in the surface layer (Wulff and Stigebrandt, 1989). The internal source term, *Intsource*, constitutes the very important difference between the improved model in Eq. (1) and the old model by Wulff and Stigebrandt (1989) where it is lacking.


The deepwater in the 100 m deep Bornholm Basin in the southern Baltic Proper switches since the 1960s between oxic and anoxic conditions. Observations from this basin confirm that the internal P source is turned on only during anoxic conditions (Stigebrandt et al. 2014). This is also supported by observations during an oxygenation experiment in the By Fjord, Sweden, where it was observed that the internal source of phosphorus from anoxic bottoms was not shut off by the first arrival of oxic

water above the sediment but later when the bottom sediment had obtained an oxidized top layer (Stigebrandt et al. 2015b). Observations in the Baltic Proper show that the top layer of the sediment was rapidly oxygenated by the recent major deepwater inflow (Rosenberg et al., 2016).

The internal sink, *Intsink = Intsink₁ + Intsink₂* (c.f. Fig. 1), can be written (Wulff and Stigebrandt, 1989)


$$Int\sin k = c_1 vA \qquad (2)$$

Here $v$ (m year$^{-1}$) is the so-called apparent settling velocity and $A$ the surface area of the Baltic Proper. The annual removal rate of phosphorus from the surface water to internal sinks is thus assumed to be proportional to the upper layer winter

concentration $c_1$. The external sink by export to Kattegat is also proportional to $c_1$. Thus all sinks are proportional to the winter surface water concentration $c_1$. From this follows that $c_1$ will increase (decrease) when the total P supply increases (decreases).



Using Eq. (2), and writing *Totsource = Extsource + Intsource + $Q_1 c_0$*, the winter surface concentration $c_1$ (mmol P m$^{-3}$) can be written as follows (from Eq. 1),

$$c_1 = \frac{Totsource - V\frac{\overline{dc}}{dt}}{vA + \gamma(Q_f + Q_1)} \qquad (3)$$

The winter surface concentration $c_1$ in the Baltic Proper is thus proportional to the total supply of phosphorus, minus the rate of change of P stored in the water column. Later in this paper we will use Eq. (3) for steady-state situations, i.e. when the storage of P in the water column does not change.

The value of the denominator in Eq. (3), which should not vary with time, is determined using data from Stigebrandt et al. (2014). In 1980, the winter concentration $c_1$ of phosphorus (TP) in the surface layer was 0.8 mmol P m$^{-3}$, *Extsource* = 60 000, *Intsource* = 46 000, $V\frac{\overline{dc}}{dt}$ = 5000 and $Q_1 c_0$ = 11 000 (tonnes P yr$^{-1}$). Inserting this in Eq. (3) one finds that $vA + \gamma(Q_f + Q_1)$ = 4520 km$^3$yr$^{-1}$. With $Q_1 \approx Q_f \approx$ 450 km$^3$yr$^{-1}$ (e.g. Stigebrandt et al., 2014) and $\gamma$=0.8 (Wulff and Stigebrandt, 1989) the flushing term $\gamma(Q_f + Q_1)$ equals 720 km$^3$yr$^{-1}$. The value of the internal sink term $vA$ thus equals 3800 km$^3$yr$^{-1}$. According to the present model the sink by export to Kattegat thus accounts for 16% (720/4520) while the internal sink accounts for 84% (3800/4520) of the total sink. This partition should be valid for all trophic steady states of the Baltic Proper. The internal phosphorus sink, which in the present case also includes P exported to the Bothnian Sea (Stigebrandt et al. 2014), should thus always be 5.3 times greater than the export sink to Kattegat.

With $A$ = 250 000 km$^2$, one finds that the apparent removal rate $v \approx$ 15 m year$^{-1}$, c.f. Eq. (2). This is twice the value estimated by Wulff and Stigebrandt (1989). The higher value found here is caused by the inclusion of the internal source in the present model. As already mentioned, this term was lacking in the model developed by Wulff and Stigebrandt (1989).

## 3 Model results and discussion

The winter surface concentration $c_1$ for steady state situations and known total P supply can be computed from Eq. (3). Present days concentrations around 1 mmol m$^{-3}$ correspond to an annual supply of about 140 000 tonnes yr$^{-1}$ (Fig. 3). It is obvious that the equilibrium concentration will be reduced if the total supply is decreased.





Sustained oxygenation of the deep bottoms of the Baltic Proper will shut off the internal source of P from anoxic bottoms (Stigebrandt et al., 2014). This means that the total P supply to the Baltic Proper can be decreased quite much from its contemporary value through oxygenation of the deepwater. Sustained oxygenation of the deepwater can thus be used as a method to restore the Baltic Proper. Should restoration occur in the nearest future, the internal source vanishes and the total P

should thus be in the range 25 000 – 40 000 tonnes yr$^{-1}$, depending on the external P supply at the time for restoration. The equilibrium winter surface concentration $c_1$ after restoration should then be in the range 0.18 – 0.29 mmol P m$^{-3}$ (Fig. 3), which is a reduction by 71-82 % as compared to the present day concentration of about 1 mmol P m$^{-3}$.

The model thus shows that provided the internal source can be shut off, by oxygenation of the bottoms, it is possible to

restore the Baltic Proper to a state in equilibrium with the external supply plus the supply from Kattegat. However, the rate of oxygen supply needed during restoration has to be computed. The existing oxygen debt, defined by the amounts of hydrogen sulfide, ammonia and other reduced substances in the water column or in contact with the water column (e.g. reduced metals), has to be paid by the supplied oxygen, see Section 4.

The rate of loading of the deepwater with fresh organic matter should be proportional to the net production of organic matter, $NP$, c.f. Fig. 2, where $NP$ is proportional to $c_1$. This means that if restoration reduces $c_1$ by 75%, the deepwater oxygen demand due to supply of fresh organic matter should be reduced by 75 %. Thus, in the course of the restoration operation, the deepwater oxygen demand is strongly reduced. This is the reason why the natural circulation of the Baltic Proper should be able to keep the deepwater oxygenated after restoration in the same manner as it did for a long period ending in the 1950s.


Hydrographical conditions in the deepwater of the Baltic Proper caused by the huge variability of the salinity of inflowing new deepwater may apparently both initiate and finish eutrophication which can be seen in deepwater sediment records. The crucial process creating eutrophication is the internal source of phosphorus that is turned on during anoxic conditions and turned off when deepwater sediments become oxygenated for a sufficiently long time whereby the system jumps to an

oligotrophic state where the trophic level is determined by the external source and the Kattegat source.

To compute the total amount of phosphorus in the water column one needs to resolve the two layers in the model. The resolved model may also be needed to make precise estimates of the time needed for the restoration. However, already after the first year with oxidized top layer of the sediments the total sink is about 130 000 and the total source is, say, 30 000

(tonnes P year$^{-1}$) whereby the storage of P in the water column will decrease by 100 000 (tonnes P year$^{-1}$) or by about 15%. With an annual storage decrease of 15%, the time needed to approach the equilibrium concentration in the surface layer can be estimated to ca 10 years. The prediction that the restoration time is of the order of one decade fits nicely with the observations of rapid termination of anoxic periods shown by Jilbert and Slomp (2013).



## 4 How to perform restoration of the Baltic Proper

In the preceding section it was shown that restoration of the Baltic Proper to a state in equilibrium with the total supply equal to the external supply plus the supply from Kattegat can be achieved in about 10 years provided the internal source from anoxic bottoms is shut off. Stigebrandt and Gustafsson (2007) suggested that oxygen for the deepwater oxygenation should

be taken from the oxygen-saturated so-called winter water, resting above the permanent halocline with oxygen content greater than 10 g $O_2$ m$^{-3}$. They also suggested that floating wind mills equipped with pumps might be used to transport winter water into the deepwater where it should be mixed with the ambient water. A similar system, powered by the electrical grid, was used in the By Fjord experiment where 2 m$^3$s$^{-1}$ of surface water was pumped into the deepwater and released through horizontal jets to obtain strong instantaneous mixing without stirring up bottoms sediments. The pumping

created downward vertical motion in the basin because the buoyant plumes lift large amounts of deepwater upwards (c.f. Fig. 2 in Stigebrandt et al. 2015b). The optimal design of pumps and their geographical localization in a pumping system for the Baltic Proper and the most efficient power supply remain to be investigated.

The amount of oxygen needed to oxidize hydrogen sulfide and ammonium in the deepwater of the Baltic Proper in the

autumn 2013, was about $2 \cdot 10^9$ and $0.5 \cdot 10^9$ kg, respectively, as calculated from hydrographical data. The debt has accumulated over several years (c.f. Fig. 1). It should be noted that oxygenation by inflow of new dense deepwater is quite inefficient. The huge inflow in December 2014 and January 2015 had in autumn 2015 made almost no effect to the oxygen debt in the water column (Fig. 1). This strongly supports the idea that only deepwater ventilation by vertical mixing, as described in Section 1, may efficiently oxygenate the deepwater when this has become anoxic. Oxygenation experiments in

lakes show that there often are huge oxygen debts in the form of organic matter in the sediments which may prolong the restoration time quite much. This should probably not be the case in the Baltic Proper where most of the bottom area has been oxygenated for long time. This conclusion is supported by the fact that earlier natural oxygenation events have been successful which should not be possible if large oxygen debts in organic matter had been present in the bottoms.

Objections against artificial oxygenation of the deepwater of the Baltic proper have been raised because of expected negative ecological effects as briefly reviewed in e.g. Stigebrandt et al. (2014). All suggested negative as well as positive effects of oxygenation should be investigated in an Environmental Impact Assessment (EIA). Work that can be included in an EIA has already been published. In the By Fjord experiment (Stigebrandt et al., 2015b) it was shown that oxygenation of the sediment did not increase the fluxes of organic and inorganic toxins from the earlier anoxic sediments. Other effects of oxygenation

are discussed in the paper by Stigebrandt et al. (2014). A model investigation of the effect of man-made oxygenation on the cod recruitment in Bornholm Basin has been performed (Stigebrandt et al., 2015a). That study shows that oxygenation of the deep waters in the Bornholm Basin should improve the hydrographical conditions required for successful cod recruitment. This is measured by the so-called cod reproduction volume (CRV defined by; $S>11$, $O_2 >2$ mL L$^{-1}$). The model computations



show that in years when the CRV was small in the Bornholm Basin under natural conditions, oxygenation would have helped to increase the CRV substantially. Keeping the deepwater oxygenated will stop leakage of phosphorus from the earlier periodically anoxic bottoms as discussed above. It will also permit colonization of the deep bottoms of the basin which will increase the food supply to e.g. cod. Furthermore, reduction of the winter surface concentration of P by 75 %

would eliminate the occurrence of large cyanobacterial blooms. A restoration system also has legal and fiscal aspects that require negotiations between the Baltic Sea states.

Restoration of the Baltic proper along the lines sketched above would in about ten years change the tropic state from the present eutrophic state to an oligotrophic state similar to that in the beginning of the 1950s. Thereafter the pumps may be

removed and the Baltic Proper should remain in the new state as long as the phosphorus supply does not increase substantially by increased external supply and/or internal supply due to development of anoxic bottoms possibly initiated by major inflows of extremely salty new deepwater. With all the experience that should be gathered during a restoration operation, it should be straightforward in future to prevent development of anoxia using oxygenation of the deepwater in critical basins when needed.

**5 Concluding remarks**

Sediment core data show that the deepwater sediments of the Baltic Proper have alternated between oxygenated and anoxic episodes. To understand these switches one must understand the water exchange of the Baltic Proper and the resulting vertical stratification of the deepwater. Due to its topographical construction with a vast horizontal area and narrow and shallow straits in the entrance area, the salinity of new deepwater varies extremely much. As a consequence of the huge

variability there may be long periods when deepwater inflows are small and less saline which means weak stratification permitting efficient oxygenation of the deepest bottoms by lowering the halocline due to wintertime convection in the surface layer. This is identified as the natural restoration mechanism, which allows sustained oxygenation of the deep bottoms breaking anoxia and eutrophication as described and modelled in the present paper. This restoration mechanism has not earlier attracted much attention.


Using a P budget model including an internal P source from anoxic bottoms, the present paper shows that by turning off the internal source, the Baltic proper may be restored in about one decade to a state in equilibrium with the total supply, which then equals the external supply plus the supply from Kattegat. The estimated short restoration time is supported by short termination time of anoxic periods as observed in sediment cores by Jilbert and Slomp (2013).


We have also briefly discussed how restoration can be achieved by man-made oxygenation of the deepwater. When the restoration is completed, the oxygenation equipment may be removed. The likelihood for a successful restoration should be

inversely proportional to the external P supply. Thanks to ambitious reductions of the external P supply since the 1980s, the present time external P supply is about the same as in the beginning of the 1950s which should facilitate restoration.

The internal supply of phosphorus from anoxic bottoms is at present about three times greater than the external supply. If nothing is done to stop the internal supply, it is generally judged that it probably may take very long time before the phosphorus concentration will decrease. To predict this time is not possible since it depends on the occurrence of the very rare situation with low-salinity mode of deepwater inflow permitting sustained oxygenation of the deepwater by halocline lowering.

## Acknowledgements

The present work was supported by the Swedish Agency for Marine and Water Management. Lars Andersson kindly supported data for 2015 for Figure 1.

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



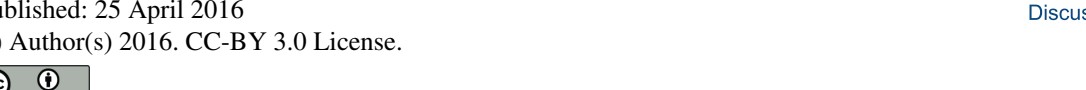

## Figures

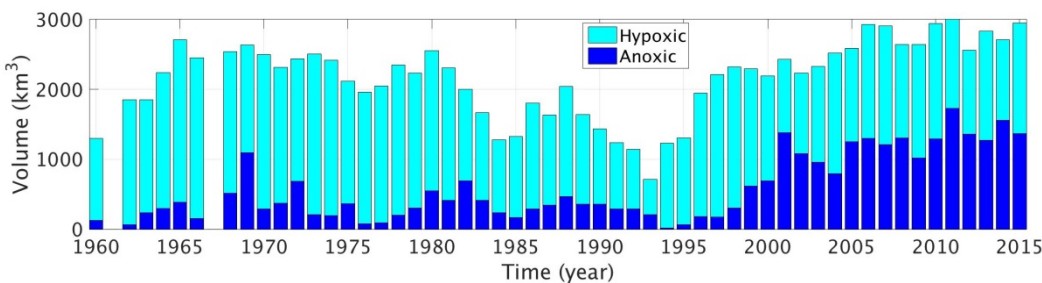

**Figure 1 Volume of hypoxic ($0<O_2<2$ mL L$^{-1}$) and anoxic (no oxygen) bottom water, observations obtained in the period August to October, in the Baltic Proper, including Gulf of Riga and Gulf of Finland, from 1960 to 2014. Results from 1961 and 1967 were omitted due to lack of data from the deep basins. (Redrawn from Hansson and Andersson (2014) and expanded by data for 2015 obtained from Lars Andersson).**





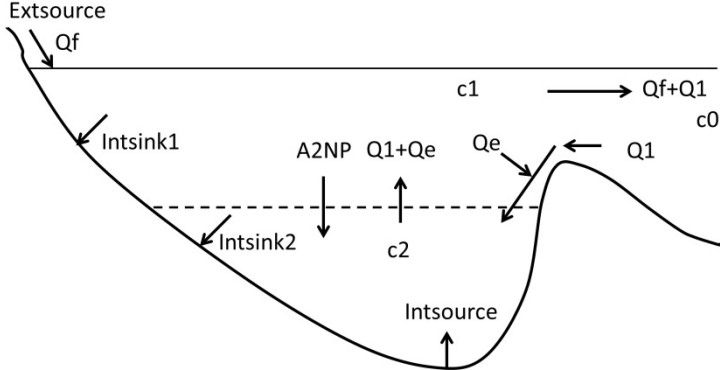

**Figure 2 Phosphorus model of a two-layered Baltic Sea with winter concentrations $c_1$ and $c_2$, in the upper and lower layers, respectively (Stigebrandt et al. 2014).The stippled line indicates the horizontal surface of area $A_2$ separating the upper and lower layers (the halocline). The P concentration, $c_0$, in Kattegat, outside the entrance sills is further discussed below Eq. (1). $Q_f$ is the rate of freshwater supply, $Q_1$ the rate of inflow of new deepwater from Kattegat and $Q_e$ the flow rate of surface water that is entrained into the inflowing new deepwater. The vertical exchange of P between the layers is due to (downward) export of new production $NP$ and surface water mixed into the new deepwater from Kattegat and (upward) transport of lower layer water into the surface layer due to entrainment in winter at the mean rate $Q_{1+}Q_e$. Internal sources and sinks associated to the seabed are also marked.**



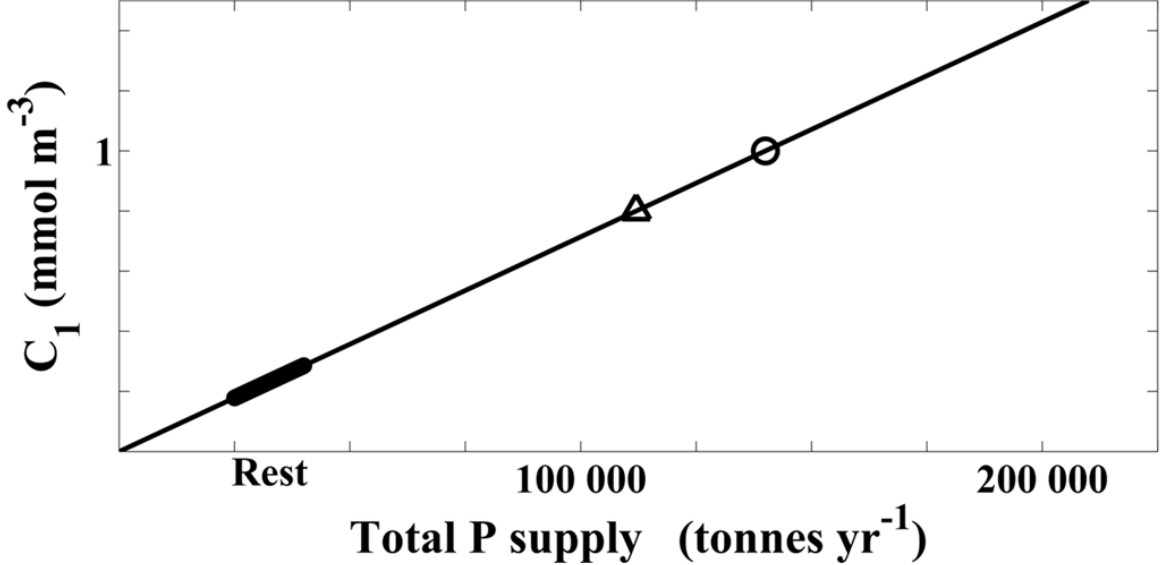

**Figure 3** The equilibrium winter surface concentration $c_1$ (TP) versus the total P supply to the Baltic proper computed using Eq. (3). The equilibrium concentration for the total P supply in 1980 (Δ) and 2005 (○) are shown together with the equilibrium concentration predicted to occur after a complete restoration (short fat line) with total P supply (i.e. the external supply plus import from Kattegat) in the range 25 000 – 40 000 tonnes yr$^{-1}$, denoted by "*Rest*".