# Peer review of "Restoration of the Baltic Proper by decadal oxygenation of the deepwater"

_Ocean Science, 2016_

## Referee Comment (RC1) · Anonymous Referee #1 · 19 May 2016

Using a conceptually simple model, the author argues that oxygenation of the deep waters of the Baltic Sea could substantially reduce the phosphorus content of the water column. The arguments presented are plausible and mostly convincing, even though there is not much new science in the manuscript. A perhaps novel aspect is the hypothesis that because of positive feedbacks the artificial oxygenation could be terminated after several years without destabilizing the restoration of oxygen and phosphorus to less eutrophic conditions. Still, all quantitative arguments rely on numbers that are just thrown in by the authors, often without reference, justification or error bars/uncertainty estimates. The model is used as convincing argumentation, but it is only an incremental advance with respect to earlier studies, including some by the same author. I value the manuscript as a summary and review paper, and thus think that there is some value in publishing it.

[Figure]

A major criticism regarding the presentation is the lack of absolute clarity required in the joint analysis of the P and O2 cycles. The manuscript begins (p.2, l.2) with the statement that "vertical circulation" transports P from the deep to the surface waters. This is needed for the positive feedback on production, export etc. However, "vertical circulation" would also transport oxygen to depth, thereby acting as a negative feedback loop, which is not mentioned in the manuscript at all. Later, it is even said that the deep box is assumed well mixed for P but stratified for O2. Again, this implies different transport agents for P and O2, which is not correct (and, I believe, not necessary for the main argument). The treatment of P and O2 transport in the argumentation and story line must be perfectly consistent. This needs careful attention in the revision.

Unfortunately, the presentation is not very convincing, some arguments are not justified, some assumptions are not stated clearly enough and some terms in the equations are not defined. Thus I cannot recommend publication of the manuscript in its present form and ask for a substantial revision addressing the individual points below. The terminology is not always precise. E.g. what do restoration (e.g. title) or recovery (e.g. p.2, l.13) mean? To have scientific value, such terms should be defined better (e.g. with respect to what reference?).

Individual points: p.3, l.6 & l.22/23. Line 3 speaks of water column, line 23 of surface concentrations. This is not the same and thus contradictory to "as already mentioned" in line 22.

p.3,l.24 This does NOT show that the model lacks a source. It could also mean that the model overestimates some sink(s).

p.4, l.2. What is the conceptual two-layer model? Does this include stratification of the deep layer?

p.4, eq.1, eq.2: explain the different Q's. Int sin k should probably read "Intsink" all in italics?

[Figure]

p.5, l.12-13. Where do the estimates for Extsource and Intsource come from? How well are they constrained? What methods has been used? What are the error bars?

p.6, l.4/5 "total P" would be a concentration (perhaps times volume), but the text uses rates.??? And why now 25000-40000 tonnes/yr and not 60000 as used one page above?

p.7 l.15 Where do the numbers come from? What do they mean? Would this the O2 that is required to make the entire deep box of the model oxic? Or does this assume some bottom boundary layer (which would be inconsistent with the P model part).

---

## Short Comment (SC1) · 25 May 2016

Thank you for an interesting paper, Prof. Stigebrandt!

I agree that artificial oxygenation of the Baltic Sea may be beneficial to cod, although the effects of reduced or removed stratification and a substantially altered vertical salinity gradient may have to be investigated further in relation to cod.

My main objection to the manuscript is that the general lack of long-term effects on P leakage of artificial oxygenation in lakes has not been addressed in the paper. P load, hypoxia/anoxia and P leakage are correlated, but the causal connections are being debated. Even if sediments are oxygenated, there may be a certain sediment depth where there is anoxia and where P can be transformed from particulate forms to dissolved forms and subsequently leak out into the water column. Thus, there may be

beneficial short-term effects from oxygenation that may be counteracted and vanish at a later point.

See, e. g.,

Conley, D. J., Bonsdorff, E., Carstensen, J., Destouni, G., Gustafsson, B. G., et al., 2009. Tackling Hypoxia in the Baltic Sea: Is Engineering a Solution? Environ. Sci. Technol., 2009, 43 (10), pp 3407–3411.

Hupfer, M., Lewandowski, J., 2008. Oxygen controls the phosphorus release from lake sediments – a long-lasting paradigm in Limnology. International Review of Hydrobiology, 93: 415–432.

Instead of oxygenation, nutrients may be abated further, and the most effective and cost effective options should in that case be applied first. For instance, it has been estimated that improved sewage treatment may reach 80% of the P target in the Baltic Sea Action Plan (and 70% of the N target).

See Hautakangas et al. (2014): http://link.springer.com/article/10.1007/s13280-013-0435-1#/page-1

Nevertheless, once the findings and conclusions in Hupfer & Lewandowski (2008) have been addressed, I support acceptance and publication of the manuscript.

Best regards, Dr Andreas Bryhn

———————————————

---

## Referee Comment (RC2) · Anonymous Referee #2 · 27 May 2016

In this paper, a simple model is used to assess whether artificial oxygenation of the Baltic Sea would lead to a major improvement of water quality. This topic is of great interest and the modeling study thus has great potential.

Unfortunately, however, the paper lacks detailed justification of the assumptions in the model, does not discuss the weaknesses of the model and many statements are made without the inclusion of any references. Furthermore, a direct comparison is made to previous periods of hypoxia but the differences between those periods and the modern period of hypoxia are not discussed. A major assumption is that sediments overlain by oxic bottom waters do not act as an internal source of P. This is not so as is also well-known from lake studies. Finally, the transient aspects of the processes in the Baltic Sea are not included in the model and the consequences of this are not discussed. I recommend major revision of the manuscript and an inclusion of a discussion of all of

these aspects.

Detailed comments.

1. p2. Lines 1-5. There are no references in this paragraph. Relevant references should be added (e.g. Conley et al., 2002 ES&T for the internal P source).

2. p2. Lines 30-34. The Carstensen et al. (2014; PNAS) study shows that enhanced respiration, which is linked to the increased external supply of P, played a key role in the expansion of the hypoxia since 1950. The text here could be modified to make this clearer.

3. P3. Lines 11-16 and further. Here, a steady state model is used whereas the Baltic Sea is in a transient state and the magnitude of the internal P source is a complex function of a large number of factors besides bottom water oxygen, including the previous depositional history of the sediment, the presence of macrofauna, the Fe-chemistry etc. That bottom water oxygenation does not shut-off P fluxes from the sediment as suggested here is well demonstrated in many lake studies (e.g. see for example the work of Katsev and Dittrich, 2013; Ecological Modeling) and also explains why high phosphate fluxes from the sediment to the overlying water are observed in many continental margin systems (see references in Ruttenberg (2003; Treatise of Geochemistry; Elsevier).

4. P4. 16-18. The immediate impact of reoxygenation (time scale of 1-2 years) can be very different from the long-term impact (>2 years to decades) because of recolonization of the sediment by fauna, saturation of the sediment Fe-oxides with P etc. This is relevant when discussing the data for the Bornholm basin and By fjord. Can the author indicate how representative the present-day Bornholm basin is for the Baltic Sea? Is the Bornholm basin recolonized by macrofauna during periods of oxic conditions that are used as a reference here? Can more detailed information on By Fjord be given? How was the P sequestered in these sediments? Was the lack of P released sustained over a period of multiple years to a decade?

5. P6. Lines 1-2. See earlier comment above. There is no evidence that sustained oxygenation will shut off the internal source of P from the sediment completely.

6. P6. Lines 4. There is no evidence that the internal source will vanish, see above.

7. P6. Line 11. The oxygen debt in the sediment (organic carbon, iron-sulfides) is very large. How would the results of the calculations change if this were included?

8. P6. Line 32. The conditions during the two previous hypoxic periods were very different: there was much less phosphate in the system and, during the hypoxic intervals of the HTM, the salinity was much higher. The author should consider these differences in the discussion in this section.

9. P7. Lines 2-4. Given the uncertainties, is it really possible to conclude that this restoration can be achieved in 10 years? The model is simple and does not account for the transient state the Baltic Sea is in and the release of P that occurs from sediments below oxic bottom waters, so is this truly justified? The oxygen debt in the sediment is also not included in the calculations (see comment above).

10. P8. Lines 2-3. See earlier comment. There is no evidence that sediment release of P will stop under oxic conditions.

11. P8. Lines 8-10. No evidence is shown that the trophic state will change from eutrophic to oligotrophic.

12. P8. Lines 28-29. See earlier comment. The modern Baltic Sea contains much more P than it did during past periods of hypoxia. There were also important other differences, such as the difference in salinity during the hypoxic interval of the HTM. They can thus not be compared directly in this manner.

---

## Author Comment (AC1) · 15 Jul 2016

os-2016-17 Authors response to Interactive comment from Anonymous Referee #1 on "Restoration of the Baltic Proper by decadal oxygenation of the deepwater" by Anders Stigebrandt. By A. Stigebrandt The author wants to thank Referee #1 for a detailed review which has led to several improvements of the presentation. Ref #1, p.1, l.9/14. .. A perhaps novel aspect is the hypothesis that because of positive feedbacks the artificial oxygenation could be terminated after several years without destabilizing the restoration of oxygen and phosphorus to less eutrophic conditions. Still, all quantitative arguments rely on numbers that are just thrown in by the authors, often without reference, justification or error bars/uncertainty estimates. ..

Author: In the introductory section the referee says that the possibility to terminate oxygenation after several years without destabilizing the restoration of oxygen and phosphorus to less eutrophic conditions might be a novel aspect of the paper. To the best of my knowledge, it is a novel aspect. Another novel aspect is the coupling of sustained oxygenation of the deep bottoms, due to occasional disappearance of the deepwater, and the rapid termination of anoxic periods as shown in sequences of sediment records. The referee also thinks that numbers in the model is just thrown in. In the revised version of my manuscript I attempt to be more pedagogic and tell how numbers are derived as described below. Ref #1, C2, l.2 – l.6: The manuscript begins with the statement that the "vertical circulation" transports P from the deep to the surface waters. This is needed for the positive feedback on production, export etc. However, vertical circulation would also transport oxygen to depth, thereby acting as a negative feed-back loop, which is not mentioned in the manuscript at all. Author: The situation described by Re #1 is valid for lakes. But the Baltic proper is also salt stratified. Usually, there is a halocline at about 60 m depth (although it sank to 100 m depth during an unusual episode in the 1980s as mentioned in the paper). The halocline rises due to inflow of new deepwater. However, on an annual basis, this is counteracted by erosion during periods of vertical convection in the surface layer in autumn and winter. Thus halocline water is transported to the surface layer by entrainment which basically is a one-way process, why oxygen from the surface layer is not transported to the deep layer by this process. In salt stratified systems, there are thus different agencies for transports of P and O2. Actually, O2 is transported to the deepwater mainly with incoming new deepwater from the entrance area (Kattegat and the Belt Sea). There is thus no negative feedback loop of the type imagined by Ref #1 as long as there is a halocline (pycnocline) separating the upper and lower layers. Manuscript changes: On p.2, l.2. To avoid misunderstanding the sentence is changed as follows. "Vertical convection in autumn and winter transports dissolved P from the deepwater to the surface layer by entrainment." Then I add the following sentence on p.2, l.5. "It should be noted that the entrainment process gives rise to one-way transports, meaning that it does not transport e.g. oxygen from the surface layer to the deepwater, which if it happened,

would counteract the positive feedback loop". Ref #1, p2, l.6 – l.10: Later it is even said that the deep box is assumed well-mixed for P but stratified for O2. Again, this implies different transport agencies for P and O2, which is not correct (and, I believe, not necessary for the main argument). The treatment of P and O2 transport in the argumentation and story line must be perfectly consistent. This needs careful attention in the revision. Author: I regret an unfortunate formulation in the manuscript that may support the interpretation made by Ref #1. The text is revised as described below. Manuscript changes: p.3, l.30 – p.4, l.1. The two sentences are replaced by the following text. "The time-dependent phosphorus model is a mass balance model that only considers sources and sinks, that all can be represented by fluxes through the external boundaries of the water mass, and storage changes in the water mass. The internal dynamics that regulate the exchange between the layers are thus not invoked in the P model." Ref #1: The terminology is not always precise. E.g. what do restoration (e.g. title) or recovery (e.g. p. 2 l. 13 mean? Author: From p.2, l.6 – l.7 in the manuscript it should be clear that restoration means that anoxic episodes are replaced by oxygenated episodes. Manuscript change: p.2, l.6, add at the end of the sentence, "due to anoxia" ; p.2, l. 13: "recovery" is replaced by "restoration". Ref #1: p.3, l.6 & 22/23: Line 3 speaks of water column, line 23 of surface concentrations. This is not the same and thus contradictory to "as already mentioned" in line 22. Author: This apparent contradictory is solved by a change in the manuscript, see below. Manuscript change: p.3, l.6. Text changed to "the winter phosphorus content in the water column, both in the 60 m deep surface layer and in the deepwater, of the Baltic proper" . . . Ref #1: p.3, l.24 This does NOT show that the model lacks a source. It could also mean that the model overestimates some sink(s). Author: Re #1 is correct. This possibility was discussed on p. 35-36 in Stigebrandt et al. (2014). Manuscript change: p. 3, l. 25. The following sentence is added. "The inverse relationship between external supply and winter surface concentration may also be explained as an effect of a decreasing P sink as discussed in Stigebrandt et al. (2014)." Ref #1: p.4, l.2: What is the conceptual two-layer model? Does this include stratification of the deep layer? Author: The conceptual twolayer model is described in Fig. 2 and the accompanying legend. A halocline separates the upper and the deep layers. This is important because it strongly hampers oxygen supply from above to the lower layer. Manuscript change: p.4, l.2: Insert "c.f. Fig. 2," after "model". Ref #1: p.4, eq. 1, eq 2: explain the different Q's. Int sin k should probable read #Intsink" all in italics. Author: The Q's and all other variables in eq. 1 and eq. 2 are defined in the legend to Fig. 2 on p.12 but that is unfortunately not mentioned in the text in connection with eq.1.Yes, Int sin k should be Intsink, but the equation editor did not allow me to write it correctly. The equation editor apparently believes that sin is the sinus function. Manuscript change: The following line is added to the sentence that ends on p.4, l.9 "$Q_f$ is the rate of freshwater supply, $Q_1$ the rate of inflow of new deepwater from Kattegat and $Q_e$ the flow rate of surface water that is entrained into the inflowing new deepwater, c.f. Fig. 2". I will change Int sin k to Intsink . Ref #1: p.5, l.12 – l.13: Where do the estimates for Extsource and Intsource come from? How well are they constrained? What methods have been used? What are the error bars? Author: As told on p. 5, l.11/12 the figures for Extsource and Intsource are adapted from Stigebrandt et al. (2014) where it is told that data for Extsource is obtained from Gustafsson et al. (2012) which is in the reference list in the manuscript (p.9, l.19/21). There is an immense collective (Helcom, other institutions) work behind the figures in Gustafsson et al. 2012 that is based on long time series of observational data and supporting modeling. This work is the most trusted estimate of the external supply that exists today. Intsource was derived by Stigebrandt et al. (2014) using the extended P model. The data input to the model is Extsource for 1980 and 2005 (from Gustafsson et al. 2012). Furthermore data for the storage change is taken from Fig. 2 in Stigebrandt et al. (2014) which is based on time series of hydrographical observations in several basins of the Baltic Proper. The P model also uses an estimate of phosphorus import with new deepwater. This is also discussed in Stigebrandt et al. (2014). Within the frame of the model used, the error in Intsource should not be very large since the errors in Extsoutce and storage changes are quite small and the estimated import from Kattegat is similar to other estimates as discussed in Stigebrandt et al. (2014). Manuscript

change: p.5, l.11/12: replace the last part of the sentence (from, is determined) with "will be derived here." P.5, l.13: Insert the following sentence: "These numbers are adopted from Stigebrandt et al. (2014) where data sources are described. Within the frame of the model used, the error in Intsource should not be very large since the errors in Extsoutce and storage changes are quite small and the estimated import from Kattegat is similar to other estimates as discussed in Stigebrandt et al. (2014)." Ref #1: p.6, l.4/5: "total P" would be a concentration (perhaps times volume), but the text uses rates. ??? And why now 25 000 – 40 000 tonnes/yr and not 60 000 as used one page above? Author: Sorry, the word "supply" is missing at the end of line 4. Thus "total P" should be "total P supply" which is a rate. The external supply was 60 000 tonnes/yr in 1980. In 2005 it was 35 000 (Gustafsson et al. 2012). The total P supply minus the internal source was 46 000 in 2005 (which includes the Kattegat source). However, the external source has continued to decrease and is at present probably less than 30 000 tonnes/yr. Should the reduction of the external source decrease until restoration is completed, which at the earliest should be 15 years from now, the external supply should certainly have decreased further to maybe 20 000 tonnes/yr. The lower range used for future external supply equals 14 000 tonnes/yr which is maybe unattainable. However, this is not important here. Manuscript change: p. 6, l.4: "total P" is changed to "total P supply". The following sentence is added on P3, l5:"Based on Gustafsson et al. (2012), Stigebrandt et al. (2014) used an external supply of 60 000 and 35 000 tonnes/yr in 1980 and 2005, respectively.". The following sentence is added on p.6, l5:"This includes the contribution from Kattegat, see above." Ref #1: p.7, l. 15: "Where do the numbers come from? What do they mean? Would this the O2 that is required to make the entire deep box of the model oxic? Or does this assume some boundary layer (which would be inconsistent with the P model part). Author: The figures are computed as follows. The amounts of H2S and NH4 in anoxic water in the autumn 2013 were computed from hydrographical data (i.e. in the water column) from several hydrographical stations in the Baltic proper and using the hypsographic volume information from the different basins. The amount of O2 needed to oxidize all H2S and NH4 in the

anoxic water were computed and presented on p.7, l.12. Stigebrandt and Gustafsson (2007) estimated that the rate of O2 supply needed to keep the (already oxygenated) Baltic proper deepwater oxic is about 3Âů109 kg yr-1). This figure was supported by the estimate in Stigebrandt et al. (2015b) who made a model study of oxygenation of the Bornholm Basin by pumping down oxygen saturated so-called winter water. The exact need of oxygen supply during a restoration operation remains to be estimated. Manuscript changes: p.7, l.6, add the following part to the sentence: "They estimated that a flux of 3Âů109 kg yr-1 would be needed to keep the deepwater oxygenated and" . . . P.7, l.11, add the following sentences: "A model pumping experiment by Stigebrandt et al. (2015) showed that the Bornholm Basin can be kept well oxygenated by pumping 1000 m3 s-1 of well oxygenated winter water into the deepwater. The exact oxygen need of a complete restoration of the Baltic proper remains to be estimated." p. 7, l.15, add the following after the word hydrographical: "and hypsographic" p.7, l.16, change (c.f. Fig. 1) to "c.f. Fig. 1 that shows the volume of anoxic water"

---

## Author Comment (AC2) · 15 Jul 2016

The author wants to thank Andreas Bryhn for his comments that led to inclusion of new text and new references in the manuscript.

Andreas Bryhn: My main objection to the manuscript is that the general lack of long-term effects on P leakage of artificial oxygenation in lakes has not been addressed in the paper. P load, hypoxia/anoxia and P leakage are correlated, but the causal connections are being debated. Even if sediments are oxygenated, there may be a certain sediment depth where there is anoxia and where P can be transformed from particulate forms to dissolved forms and subsequently leak out into the water column. Thus, there may beneficial short-term effects from oxygenation that may be counter-acted and vanish at a later point. See, e. g., Conley, D. J., Bonsdorff, E., Carstensen,

J., Destouni, G., Gustafsson, B. G., et al., 2009. Tackling Hypoxia in the Baltic Sea: Is Engineering a Solution? Environ. Sci. Technol., 2009, 43 (10), pp 3407–3411. Hupfer, M., Lewandowski, J., 2008. Oxygen controls the phosphorus release from lake sediments – a long-lasting paradigm in Limnology. International Review of Hydrobiology, 93: 415–432.

Author: Possible long-term effects regarding the internal P load are now mentioned as one of the remaining important tasks for a complete EIA for restoration of the Baltic Sea by oxygenation of the deepwater.

Manuscript changes: p.7., l.26: replace "as briefly reviewed" with "e.g. "Conley et al. (2009) and the brief review".

p.8., l. 2/4, The two sentences are rewritten as follows: "Keeping the deepwater oxygenated will permit colonization of the deep bottoms of the basin which will increase the food supply to e.g. cod. It will also stop leakage of phosphorus from the earlier periodically anoxic bottoms as discussed above." After this the following text is added. " An important task for the EIA is to investigate if increasing leakage of P from sediments may be a long-term effect like it has turned out to be in some artificially oxygenated lakes (Hupfer and Lewandowski, 2008; Katsev and Dittrich, 2013). However, it should be stressed that nobody has shown that lake experience of P leakage is applicable to marine environments with deepwater salinity >10 as in e.g. the Baltic proper."

p.9, l. 19: the following reference is added: "Conley, D. J., Bonsdorff, E., Carstensen, J., Destouni, G., Gustafsson, B. G., et al., 2009. Tackling Hypoxia in the Baltic Sea: Is Engineering a Solution? Environ. Sci.Technol., 2009, 43 (10), pp 3407–3411."

Andreas Bryhn: Instead of oxygenation, nutrients may be abated further, and the most effective and cost effective options should in that case be applied first. For instance, it has been estimated that improved sewage treatment may reach 80% of the P target in the Baltic Sea Action Plan (and 70% of the N target). See Hautakangas et al. (2014): http://link.springer.com/article/10.1007/s13280-013- 0435-1#/page-1.

Author: I appreciate but do not share your opinion in this matter. Experience from the last 40 years has shown the eutrophication has increased in spite of large reductions of the external P supply. We know that an increasing internal P supply is the reason for this. The logical conclusion would be that one should try to decrease the internal supply. In my manuscript I suggest that this may be done by oxygenation of the deep bottoms. It is certainly true that the probability of getting long lasting results of a restoration will be higher the smaller the external load is. Thus, in my opinion one should do both oxygenation and decrease the external supply of P.

Manuscript: no change is undertaken.

---

## Author Comment (AC3) · 15 Jul 2016

The author wants to thank Referee #2 for a detailed review with many questions about the validity of the model. This has led to several improvements of argumentation and presentation.

Ref #2, p1, l9 – p2, l1. Unfortunately, however, the paper lacks detailed justification of the assumptions in the model, does not discuss the weaknesses of the model and many statements are made without the inclusion of any references. Furthermore, a direct comparison is made to previous periods of hypoxia but the differences between those periods and the modern period of hypoxia are not discussed. A major assumption is that sediments overlain by oxic bottom waters do not act as an internal source of P. This is not so as is also well known from lake studies. Finally, the transient aspects of

the processes in the Baltic Sea are not included in the model and the consequences of this are not discussed. I recommend major revision of the manuscript and an inclusion of a discussion of all of these aspects.

Author: Application of a transient mass balance model for P on the Baltic Proper and data on P release from sediments in the Bornholm Basin presented in Stigebrandt et al. (2014) strongly support the hypothesis that oxygen controls the P release from sediments in the Baltic Proper. In situ observations of benthic fluxes reported by Viktorsson et al. (2013) support the model study. Further support was given by Stigebrandt et al. (2014) who presented (their Fig. 3) a striking correlation between the area of anoxic bottoms and the P content in the water mass below 60 m depth in the Baltic Proper. Conley et al. (2002) showed a similar correlation and estimated fluxes from an internal source and Gustafsson and Stigebrandt (2007) estimated the release of P as single doses when bottoms become anoxic. From the extensive evidence presented here, there is no doubt about the oxygen control of P release from sediments in the Baltic proper in its present state. This was not clearly pointed out in the manuscript but will be so, see below. It is clear that the oxygen control of P release from sediments is quite different in different kinds of lakes. Other factors than oxygen may be of greater importance. According to Hupfer and Lewandowski (2008) more than half of North American lakes exhibited an extremely low level of P release from sediment even after the onset of anoxia in the hypolimnion despite their variability in water chemistry, trophic states and geographical locations. These authors also mention additional cases of none or delayed P release under anoxic conditions. An important component of the review of Re #2 rests on an assumed similarity between sedimentary P dynamics in the Baltic Sea and lakes. However, nobody has shown that experience from lakes is valid for the Baltic Proper. Due to several crucial differences like salt in the water, and permanent salinity stratification that is not broken down during autumn/winter, and marine instead of limnic ecology one may expect that different P burial processes are important in the Baltic Sea and in freshwater lakes, why comparisons with freshwater lakes might be inappropriate. Manuscript changes: p.2, l. 16. The following text will be inserted. "Ap-
plication of a transient mass balance model for P on the Baltic Proper and data on P release from sediments in the Bornholm Basin presented in Stigebrandt et al. (2014) strongly support the hypothesis that oxygen controls the P release from sediments in the Baltic Proper. In situ observations of benthic fluxes reported by Viktorsson et al. (2013) support the model study. Further support was given by Stigebrandt et al. (2014) who showed (their Fig. 3) a striking correlation between the area of anoxic bottoms and the P content in the water mass below 60 m depth in the Baltic Proper. Conley et al. (2002) showed a similar correlation and estimated fluxes from the internal source and Gustafsson and Stigebrandt (2007) estimated the release of P as single doses when bottoms become anoxic. From the evidence presented here there should be no doubt about the oxygen control of P release from sediments in the Baltic proper in its present state." p.9, l.18: the following reference will be inserted: "Conley, D.J., Humborg, C., Rahm, L, Savchuk, O.P., and Wulff, F., Hypoxia in the Baltic Sea and basin-scale changes in phosphorus biogeochemistry. Environmental Science and Technology 36: 5315 – 5320. DOI: 10.1021/Es025763w, 2002." p.9, l.23: the following reference will be inserted: " Gustafsson, B.G. and Stigebrandt, A.: Dynamics of nutrients and oxygen/hydrogen sulfide in the Baltic Sea deep water. Journal of Geophysical Research – Biogeosciences 112: G02023. DOI: 10.1029/2006jg000304, 2007." p.9, l.23: the following reference will be inserted: "Hupfer, M., and Lewandowski, J.: Oxygen controls the phosphorus release from lake sediments – a long-lasting paradigm in limnology. Internat. Rev. Hydrobiol., 93: 415-432, 2008." p.9, l. 21: the following reference will be inserted: "Viktorsson, L., Ekeroth, N., Nilsson, M., Kononets, M, and Hall, P.O.J.: Phosphorus recycling in sediments of the central Baltic Sea. Biogeosciences 10: 3901-3916. DOI: 10.5194/bg-10-3901-2013, 2013" Detailed comments. Re #2:1. p2. Lines 1-5. There are no references in this paragraph. Relevant references should be added (e.g. Conley et al., 2002 ES&T for the internal P source).

Author: I agree. Two references will be added.

Manuscript changes: p.2, l.2 the following text is added after "increases": "(Conley et

al., 2002; Stigebrandt et al., 2014)".

Re #2:2. p2. Lines 30-34. The Carstensen et al. (2014; PNAS) study shows that enhanced respiration, which is linked to the increased external supply of P, played a key role in the expansion of the hypoxia since 1950. The text here could be modified to make this clearer.

Author: Text modified, see below.

Manuscript changes: P. 2, l. 32. The following text is inserted between "increased" and "external": "respiration linked to increased". Insert "(Carstensen et al. 2014)" the end of this sentence.

Re #2: 3. P3. Lines 11-16 and further. Here, a steady state model is used whereas the Baltic Sea is in a transient state and the magnitude of the internal P source is a complex function of a large number of factors besides bottom water oxygen, including the previous depositional history of the sediment, the presence of macrofauna, the Fe-chemistry etc. That bottom water oxygenation does not shut-off P fluxes from the sediment as suggested here is well demonstrated in many lake studies (e.g. see for example the work of Katsev and Dittrich, 2013; Ecological Modeling) and also explains why high phosphate fluxes from the sediment to the overlying water are observed in many continental margin systems (see references in Ruttenberg (2003; Treatise of Geochemistry; Elsevier).

Author: Re #2 is wrong about steady state. It should be very clear both from the text on p. 4, l. 3 and the left hand side of Equation (1) that the model is time-dependent when used to estimate the internal source, see also the application of the model in Stigebrandt et al. (2014). I think that it is irrelevant to refer to lake studies unless the studies concern lakes that are similar to the Baltic proper, e.g. they must have a clear oxygen control on P sediment fluxes like the Baltic proper. The influence of macrofauna was discussed in Stigebrandt et al. (2015b). There is an apparent flux from sediments even under oxic conditions as shown for the Bornholm Basin by Stigebrandt et al.

[Figure]

(2014).But this flux is likely connected to decomposition of fresh organic matter and not part of the internal source.

Manuscript changes: p. 3, l. 12, "time-dependent" is inserted for clarity. The following text will be added at the end of p.6 (just before chapter 4. "It is often argued that bottom water oxygenation does not shut off P fluxes from sediments because this has been seen in many lake studies. However, nobody has proven that experience from lakes is valid for the Baltic Proper. Due to several crucial differences like salt in the water, and permanent salinity stratification that is not broken down during autumn/winter, and marine instead of limnic ecology one may expect that important P burial processes might be different in the Baltic Sea and in freshwater lakes why comparisons with freshwater lakes might be considered inappropriate until the opposite has been proven." Manuscript change. The change in response to the interactive comment by Andreas Bryhn is also relevant as an answer to the present comment by Re #2. The two sentences on p.8, l. 2 – l.4, are rewritten as follows: "Keeping the deepwater oxygenated will permit colonization of the deep bottoms of the basin which will increase the food supply to e.g. cod. It will also stop leakage of phosphorus from the earlier periodically anoxic bottoms as discussed above." After this the following text is added. " An important task for the EIA is to investigate if increasing leakage of P from sediments may be a long-term effect like it has turned out to be in some artificially oxygenated lakes (Hupfer and Lewandowski, 2008; Katsev and Dittrich, 2013). However, it should be stressed that nobody has shown that lake experience of P leakage is applicable to marine environments with deepwater salinity >10 as in e.g. the Baltic proper."

Re #2: 4. P4. 16-18. The immediate impact of reoxygenation (time scale of 1-2 years) can be very different from the long-term impact (>2 years to decades) because of recolonization of the sediment by fauna, saturation of the sediment Fe-oxides with P etc. This is relevant when discussing the data for the Bornholm basin and By fjord. Can the author indicate how representative the present-day Bornholm basin is for the Baltic Sea? Is the Bornholm basin recolonized by macrofauna during periods of oxic

conditions that are used as a reference here? Can more detailed information on By Fjord be given? How was the P sequestered in these sediments? Was the lack of P released sustained over a period of multiple years to a decade?

Author: Long-term effects regarding the internal source were discussed for the Bornholm Basin in Stigebrandt et al. (2014). It was estimated that the total loss estimated from hydrographic observations in the basin since the 1960s should have used about 20 % of the P stored in the upper 20 cm of the sediment. It was also found that the P supply to the bottom water during oxic conditions was greater than the P supply by decomposition of fresh organic matter as estimated from oxygen consumption. However, it was concluded that the estimates for oxic periods were not sharp enough to exclude contributions from fluxes from anoxic sediments. Effects of recolonization of bottom sediments by fauna in the Bornholm Basin were discussed in Stigebrandt et al. (2015b) who also referred to other published studies in the Baltic proper. It is believed that the Bornholm Basin is representative for the Baltic proper with respect to oxygen control of P release from sediments. Re #2 is right that we do not know answers to all questions regarding long-term effects of oxygenation on P fluxes why this should be an area of continued research.

The long-term impact of oxygenation has not been studied in the By Fjord that cannot be restored in the same way as the Baltic proper. This is because anoxia in the deepwater of the By Fjord is not due to local production driven by internal P load. Anoxia in the deepwater is mainly due to large import of organic matter over the entrance sill in combination with very weak vertical mixing and large long-term components in the variability of density in the coastal water leading to very long residence time for the deepwater. As expected, a few years after the termination of the oxygenation of the deepwater, the By Fjord had returned to a state similar to that it had when the oxygenation started. No study of P in the sediment was undertaken in the By Fjord. A lot of efforts were spent to in situ observations of benthic fluxes using benthic lander chambers; see Stigebrandt et al. (2015a) but the time series were ended shortly after

the end of the pumping.

Manuscript changes: p.4, l. 22, the following sentence is added. "Long-term effects of oxygenation on P fluxes have not been studied, e.g. effects of fauna, but were discussed by Stigebrandt et al. (2015b)."

Re #2: 5. P6. Lines 1-2. See earlier comment above. There is no evidence that sustained oxygenation will shut off the internal source of P from the sediment completely.

Author: As discussed earlier, the observations from Bornholm Basin show that there is a flux even during oxic conditions but most, if not all, of this flux is due to decomposition of fresh organic matter.

Manuscript changes: The text added to p. 8 in response to the detailed comment no. 3 covers this aspect.

Re #2: 6. P6. Lines 4. There is no evidence that the internal source will vanish, see above.

Author: I think there is, see the new paragraph above that will be included on p. 2, l.16

Manuscript changes: None.

Re #2: 7. P6. Line 11. The oxygen debt in the sediment (organic carbon, iron-sulfides) is very large. How would the results of the calculations change if this were included?

Author: Oxygen debts in the water mass and in the sediment influence the time it takes to stop the internal P source and of course, the amount of O2 needed to restore. Substances in the sediment that were oxidized during and after water renewal and reduced when the water run out of oxygen were included in the model by Stigebrandt et al. (2015b). The existence of these substances was clearly seen in the results as discussed in that paper. They delay the arrival of anoxia during stagnation and consume some of the oxygen brought in during water renewal. These substances modify but do not change the nature of the response to oxygenation.

Manuscript changes: p.7, l. 11. The following sentence is inserted: "This debt plus a debt due to reduced substances in the sediment were included in the model for the Bornholm Basin by Stigebrandt (2015b). It was found that the sediment debt slightly modified the response to oxygenation."

Re #2: 8. P6. Line 32. The conditions during the two previous hypoxic periods were very different: there was much less phosphate in the system and, during the hypoxic intervals of the HTM, the salinity was much higher. The author should consider these differences in the discussion in this section.

Author: A comment on this will be included in the manuscript, see below.

Manuscript changes: p. 8, l. 33, add the following text: "The similarity between the Baltic Sea of today and during the anoxic period occurring under the Medieval Climate Anomaly should be quite large but the similarity is less for earlier anoxic periods due to topographic differences caused by sea level changes and relative land rise, c.f. Jilbert et al. (2015). p.9, l.26, add reference: "Jilbert, T., Conley, D.J., Gustafsson, B.G., Funkey, C.P., and Slomp, C.P.: Glacio-isostatic control on hypoxia in a high-latitude shelf basin. Geology. Doi: 10.1130/G36454.1 2015.

Re #2: 9. P7. Lines 2-4. Given the uncertainties, is it really possible to conclude that this restoration can be achieved in 10 years? The model is simple and does not account for the transient state the Baltic Sea is in and the release of P that occurs from sediments below oxic bottom waters, so is this truly justified? The oxygen debt in the sediment is also not included in the calculations (see comment above).

Author: The model is not as simple as described by Re #2. As told before the model is transient and the internal source has been estimated using the transient model. Actually, oxygen debts will prolong the restoration time and they influence the amount of oxygen needed to perform the restoration. A qualified estimate is that the present debts, given on p. 7, l. 15, are of the same magnitude as the oxygen supply during one year that has been discussed for a restoration system. The debts can therefore be

**OSD**

[Figure]

estimated to prolong the restoration by one year.

Manuscript changes: No

Re #2: 10. P8. Lines 2-3. See earlier comment. There is no evidence that sediment release of P will stop under oxic conditions.

Author: I think there is strong evidence as described in the new text that will be inserted on p3, l. 16 (see above).

Manuscript changes: No

Re #2: 11. P8. Lines 8-10. No evidence is shown that the trophic state will change from eutrophic to oligotrophic.

Author: This is a model result described in the beginning of Chapter 3 "Model results and discussion", see also Fig. 3. Restoration means that the internal source (about 100 000 tonnes P yr-1) is stopped so that the total P supply decreases from 140 000 to 40 000. The model predicts that the tot P winter surface concentration will decrease from the present concentration of about 1 to a much smaller value in the interval 0.2 – 0.3 (mmol m-3) which I believe means a change from eutrophic to oligotrophic.

Manuscript changes: No.

Re #2:12. P8. Lines 28-29. See earlier comment. The modern Baltic Sea contains much more P than it did during past periods of hypoxia. There were also important other differences, such as the difference in salinity during the hypoxic interval of the HTM. They can thus not be compared directly in this manner.

Author: The text added to the manuscript as a result of the detailed comment no. 8 covers this case also. Manuscript changes: No.